# Deranged Coagulation Profile Secondary to Cefazolin Use: Case Report

**Jinghao Nicholas Ngiam** [1] , **Tze Sian Liong** [1], **Sai Meng Tham** [2], **Thanawin Pramotedham** [1], **Rawan AlAgha** [2], **Joy Yong** [3], **Paul Anantharajah Tambyah** [2,4] **and Lionel Hon Wai Lum** [2,*]

1   Department of Medicine, National University Health System, Singapore 119228, Singapore; nicholas_ngiam@nuhs.edu.sg (J.N.N.); tzesian.liong@mohh.com.sg (T.S.L.); thanawin.pramotedham@mohh.com.sg (T.P.)
2   Division of Infectious Diseases, Department of Medicine, National University Health System, Singapore 119228, Singapore; sai_meng_tham@nuhs.edu.sg (S.M.T.); Rawan_Almuataz@nuhs.edu.sg (R.A.); mdcpat@nus.edu.sg (P.A.T.)
3   Department of Pharmacy, National University Health System, Singapore 119228, Singapore; joy_yong@nuhs.edu.sg
4   Yong Loo Lin School of Medicine, National University of Singapore, Singapore 119228, Singapore
*   Correspondence: lionel_lum@nuhs.edu.sg; Tel.: +(65)-67723596; Fax: +(65)-67794112

**Abstract:** Cefazolin is a widely used first-generation cephalosporin. While generally well tolerated, several case reports have described severe coagulopathy induced by intravenous (IV) cefazolin. This was seen particularly in patients with impaired renal function, where antibiotic choice is limited and may require specific dose adjustments. Altered renal handling of antibiotics and their metabolites may potentiate toxicity and side effects. We report a case of a 72-year-old Chinese man who had been treated for methicillin-sensitive staphylococcus aureus (MSSA, coagulase-positive) infective endocarditis with cefazolin and, consequently, developed significantly elevated international normalised ratio (INR) while on therapy. This resolved within 48 h after cessation of cefazolin and administration of oral vitamin K. Malnourished patients with pre-existing or acute kidney injury may be at an increased risk of cefazolin-related coagulopathy.

**Keywords:** cefazolin; coagulopathy; elevated international normalised ratio





## 1. Introduction

Cefazolin is a widely used first-generation cephalosporin. While generally well tolerated, several case reports have described severe coagulopathy induced by intravenous (IV) cefazolin [1,2]. Particularly in patients with impaired renal function, antibiotic choice is limited and may require specific dose adjustments. Altered renal handling of antibiotics and their metabolites may potentiate toxicity and side effects [1–3].

We report a case of a 72-year-old Chinese man who had been treated for methicillin-sensitive staphylococcus aureus (MSSA, coagulase-positive) infective endocarditis with cefazolin and, consequently, developed significantly elevated international normalised ratio (INR) while on therapy.

## 2. Case Report

A 72-year-old Chinese man with hypertension, end-stage kidney disease, as well as a prior bioprosthetic aortic valve replacement for severe aortic stenosis had been admitted to the intensive care unit (ICU) with septic shock. Prior to admission, he had been on oral antihypertensive agents but had not been not on any antiplatelets or anticoagulation. There had been no new medications or newly initiated supplements. Three consecutive blood cultures revealed MSSA bacteraemia and transoesophageal echocardiography confirmed prosthetic valve infective endocarditis. The tunnelled dialysis catheter was removed, and IV cefazolin 1 g 12-hourly was started. His fever lysed, blood pressure stabilised, and

repeat blood cultures at 72 h were negative for bacterial growth. The patient did not receive anticoagulation as an inpatient and was initiated on peritoneal dialysis during his hospital stay. Intravenous heparin was not administered during dialysis.

His total white cell count improved from 19.09 ($\times 10^9$/L) to 12.63 ($\times 10^9$/L) by day 14 of admission, while his haemoglobin concentration remained stable between 7–8 g/dL. His platelet count was low at presentation 14 ($\times 10^9$/L) but rose to 225 ($\times 10^9$/L) by day 14. His baseline INR was 0.93, which rose slightly to 1.39 on day 8, and then sharply to 8.24 on day 14 of his admission (Figure 1). While normal at presentation, his PT was 73.4 s, and activated partial thromboplastin time (aPTT) was 88.0 s on day 14. His fibrinogen (4.81 g/L) and d-dimer (8.0 µg/mL) were both elevated. There was no evidence of worsening sepsis, bleeding, liver failure, or elevated transaminases.

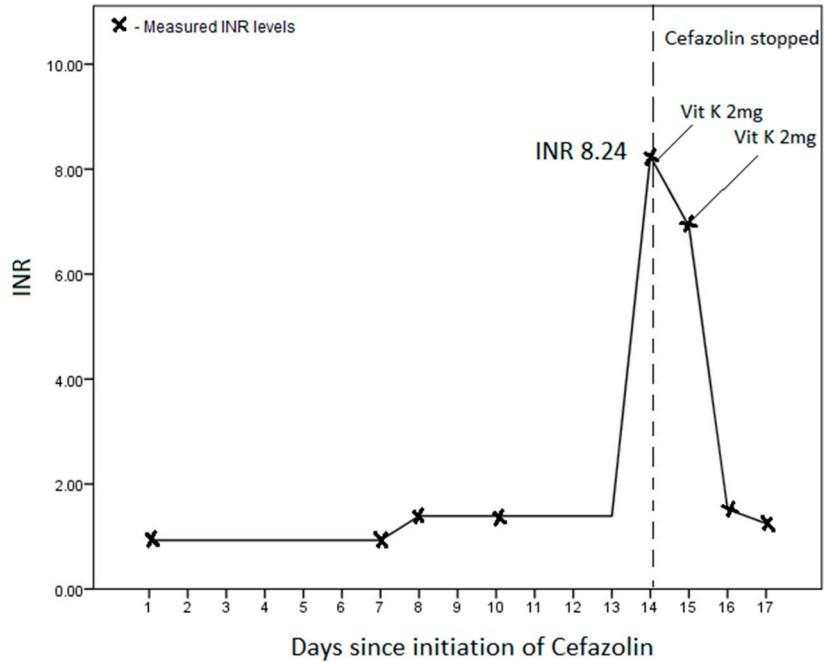

**Figure 1.** International normalised ratio (INR) trend on initiation and subsequent cessation of IV cefazolin.

We did not measure serum cefazolin levels during therapy. However, in the absence of an alternative reason for the markedly elevated INR, cefazolin was implicated and stopped. A total of four milligrams of oral vitamin K was given over the next two days, which led to a rapid improvement in the INR (Figure 1). After three days of stopping the cefazolin, the patient had two episodes of melaena, which was managed conservatively with IV esomeprazole. He had persistent hypotension and was unable to continue with renal replacement therapy and, subsequently, transited to palliative care.

## 3. Discussion

In prior case reports, cefazolin use has been demonstrated to induce vitamin-K-deficient coagulopathy [1–6]. Although more commonly associated with ticarcillin and nitrofurantoin, this remained a less well-known complication of cefazolin [1]. While the underlying mechanism was not fully understood, several mechanisms have been proposed. It may be the result of altered intestinal flora due to the antimicrobial effect of cefazolin, which is the main source of vitamin K [2]. Alternatively, the cephalosporin itself may exert a direct effect in suppressing synthesis of vitamin K by the liver [2,3]. Structurally, cefazolin contains the methyl-thiadiazole side chain that appeared to exert similar effects to vitamin K epoxide reductase inhibitors. Other cephalosporins without this side chain do not exert

similar effects [4]. The result is a patient who has prolonged PT, aPTT, and elevated INR with an increased bleeding risk.

Several factors may predispose patients to developing cefazolin-induced coagulopathy. First, a component of malnutrition, with a reduced oral intake of vitamin K, may be contributory. Of note, our patient had a low serum albumin of 17 g/dL, and hypoalbuminemia had been reported in several cases of cefazolin-induced coagulopathy [4,6]. Second, the effect appears to be dose-dependent. Patients with higher doses of cefazolin or with more prolonged courses of treatment tended to develop more severe coagulopathy (Table 1).

**Table 1.** Summary of prior case reports on cefazolin-related deranged coagulopathy.

| Study and Year | Age/ Sex | Comorbidities | Treatment Indication | Dose of IV Cefazolin | Duration of Treatment to Onset of Coagulopathy | Peak Deranged Coagulation | Time to Resolution After Cessation of Drug | Complications |
|---|---|---|---|---|---|---|---|---|
| Our study | 72/M | End-stage renal failure | S. aureus infective endocarditis | 1 g 12 h | 14 days | INR 8.29 | 2 days | Gastrointestinal bleeding |
| Chung et al., 2008 [1] | 50/F | Acute renal failure | E. coli bacteraemia | 1 g q24 h | 7 days | INR 4.0 | 2 days | None |
| Shaikh et al., 2013 [2] | 63/M | End-stage renal failure, hypertension, diabetes mellitus | Surgical prophylaxis | - | 4 days | INR 4.2 | 2 days | None |
| Kurz et al., 1986 [4] | 26/M | Acute renal failure following rhabdomyolysis | Surgical prophylaxis | 2 g 12 h | 12 days | Normotest 17% | 2 days | Surgical wound bleeding |
| Shimada et al., 1984 [5] | 79/F 87/F 71/F 87/F 74/M | - - - - - | Pneumonia Colitis Pyelonephritis Pneumonia, UTI Pneumonia | 3 g daily 2 g daily 2 g daily 3 g daily 2 g daily | 21 days 2 days 20 days 15 days 2 days | PT 65.4 s PT 15.0 s PT 21.3 s PT 27.7 s PT 28.2 s | All cases: 1–2 days | Bruising, Corneal bleeding None Haematuria Haematuria Haematemesis/ Haemoptysis |
| Kuypers et al., 2002 [6] | 45 | End-stage renal failure | Surgical prophylaxis for cadaveric renal transplant | 2 g for 3 doses Then 6 g daily for 3 days | 5 days | PT 74.6s, Normotest <10% | | Intracranial bleeding |

In addition to the above factors, abnormal metabolism and excretory function in patients with impaired renal function may also contribute to cefazolin toxicity [3,7]. True enough, this phenomenon appeared to be frequently reported in patients with acute kidney injury or end-stage kidney disease [1,2,4,5,7,8]. Patients with uraemia, in addition to being malnourished, may also demonstrate reduced renal clearance of the drug, which thereby potentiates its toxicity [3,7].

While on therapy, the onset of coagulopathy appeared to be variable, from as early as 2–4 days into therapy [2,6] in some patients, and up to 14–21 days in others. Upon cessation of cefazolin and commencement of vitamin K therapy, the coagulopathy consistently resolved within 48 h [1,2,5–7]. Although some patients did not experience any complications, others had bleeding manifestations, which may have been mild (bruises, haematuria) or more severe (gastrointestinal bleeding, intracranial haemorrhage) [5–7].

The prevalence of cefazolin-induced coagulopathy remains unknown [9], and it is not routine to monitor the coagulation profile while on cefazolin [10]. Cefazolin levels are also not routinely measured. While larger prospective studies are required to examine this association, we suggest that monitoring of the coagulation profile may be considered in patients on prolonged therapy, and in particular, patients who are malnourished or

have renal impairment. Furthermore, in these "high-risk" patients, monitoring of cefazolin levels may also be useful to avoid toxicity [11].

**Author Contributions:** J.N.N., T.S.L., S.M.T., and T.P. were involved in the conception, data interpretation, literature review, and writing of the manuscript. R.A., J.Y., P.A.T., and L.H.W.L. were involved in the conception, writing, and critical review of the manuscript. All authors have read and agreed to the published version of the manuscript.

**Funding:** This research received no external funding.

**Institutional Review Board Statement:** The study was conducted according to the guidelines of the Declaration of Helsinki, and presented fully anonymised non-identifiable information of a single-patient case report that did not require prior approval by the Institutional Review Board.

**Informed Consent Statement:** Written informed consent was obtained from the patient involved in the study.

**Data Availability Statement:** Data may be made available on reasonable request from the corresponding author.

**Conflicts of Interest:** The authors declare no conflict of interest.

**Ethics/Patient Consent:** Written informed consent was obtained from the patient prior for this report.

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
