# Peer review of "Deranged Coagulation Profile Secondary to Cefazolin Use: Case Report"

_2036-7449, doi:10.3390/idr13010021_

Round 1

Reviewer 1 Report

Did authors evaluate the pharmacokinetics of cefazolin? In other words was cefazolin levels in plasma measured?

Authors need to refer to abnormal metabolic or excretory function as an additional parameter for cefazolin related complications (1. Anon. Coagulopathy with Cefazolin in Uremia. http://dx.doi.org/101056/NEJM197406062902317 2010. Available at: https://www.nejm.org/doi/10.1056/NEJM197406062902317. Accessed February 8, 2021.)

 Did authors rule out other possibilities/confounding variables such as hemodialysis without heparin or using deltaparin with a normal factor X?

Was MSSA coagulase negative or positive?

Regarding INR graph, more explanation regarding sampling and doses is needed. Your sampling procedure must be one data point per day; however, your graph shows a continuous line which is incorrect.

Author Response

Manuscript Number: idr-1111150

Title: Deranged coagulation profile secondary to cefazolin use: case report

We thank the Editor for allowing us the opportunity to revise our manuscript and the Reviewers for the detailed and constructive comments. We have amended our paper in order to address important points raised by the Reviewers.

In the sections below, each of the points raised is identified and addressed with changes in the revised manuscript.

Reviewer 1:

Did authors evaluate the pharmacokinetics of cefazolin? In other words was cefazolin levels in plasma measured?

We thank the reviewer for this comment. Regretfully, our hospital laboratory did not have the capability to measure cefazolin levels, and it is also not routinely measured. We have added this to the description of the case report for greater clarity (Page 2).

Authors need to refer to abnormal metabolic or excretory function as an additional parameter for cefazolin related complications (1. Anon. Coagulopathy with Cefazolin in Uremia. http://dx.doi.org/101056/NEJM197406062902317 2010. Available at: https://www.nejm.org/doi/10.1056/NEJM197406062902317. Accessed February 8, 2021.)

We thank the reviewer for this comment. Indeed, it is an important aspect of the metabolism of cefazolin that leads to the complications observed. We have expanded on our discussion to include these points (Page 3).

 Did authors rule out other possibilities/confounding variables such as hemodialysis without heparin or using deltaparin with a normal factor X?

We thank the reviewer for this insightful comment. The patient had the offending tunnelled dialysis catheter removed at the start of the admission. He was then initiated on peritoneal dialysis during the admission. His peritoneal dialysis did not require any anticoagulation or intravenous heparin. Therefore, the dialysis was not likely to be contributory to the coagulopathy observed. We have added these important details to case report (Page 2).

Was MSSA coagulase negative or positive?

The MSSA was coagulase positive. We have added this to our case report (Page 1).

Regarding INR graph, more explanation regarding sampling and doses is needed. Your sampling procedure must be one data point per day; however, your graph shows a continuous line which is incorrect.

We thank the reviewer for this comment. We have revised the INR graph to mark out the points in which sampling for INR had been done. We hope this now adds clarity to the figure in portraying the INR trend while on cefazolin (Page 2).

We thank the Reviewers and the Editor for the kind and helpful comments. We hope the paper is now suitable for publication in the Journal.

Thank you!

Dr Nicholas Ngiam

Dr Lionel Lum

Reviewer 2 Report

Well-written and interesting work. As long as this is presented as a case report, the CARE guideline must be taken into account.

Additionally, information about the starting dosage of Cefazolin and clarification about patient co-morbidities or recent pharmaceutical history should be added.

The bibliography, also, is rather limited. 

Author Response

Manuscript Number: idr-1111150

Title: Deranged coagulation profile secondary to cefazolin use: case report

We thank the Editor for allowing us the opportunity to revise our manuscript and the Reviewers for the detailed and constructive comments. We have amended our paper in order to address important points raised by the Reviewers.

In the sections below, each of the points raised is identified and addressed with changes in the revised manuscript.

Reviewer 2:

Well-written and interesting work. As long as this is presented as a case report, the CARE guideline must be taken into account.

We thank the Reviewer for this comment. We have reviewed the CARE guideline and checklist and ensured that our case report complies with this format. We have changed the title to include the words “case report” as per the CARE guidelines.

We ensure that the rest of the aspects of the checklist including the title, key words, abstract, introduction, patient information, clinical findings, diagnostic assessment, therapeutic intervention, follow-up and outcomes, and discussion are in accordance with the guidelines. We also affirm that informed consent was provided by the patient for the writing of this case report.

Additionally, information about the starting dosage of Cefazolin and clarification about patient co-morbidities or recent pharmaceutical history should be added.

We thank the reviewer for this comment. We have now included our starting dosage of cefazolin (1g 12h), and details of the patient’s comorbidities (hypertension, end-stage renal disease, bioprosthetic aortic valve replacement) in the case report (Page 1)

The bibliography, also, is rather limited.

We thank the Reviewer for this comment and have since expanded on our bibliography/reference list to include other relevant and important references (Page 5).

We thank the Reviewers and the Editor for the kind and helpful comments. We hope the paper is now suitable for publication in the Journal.

Thank you!

Dr Nicholas Ngiam

Dr Lionel Lum